# OpenReview forum: "Precision-Induced Miscalibration: Understanding and Correcting Confidence Distortion in Quantized Neural Networks"
_ICML.cc/2026/Conference — ICML 2026 regular_

### Official Review · Reviewer_HW5p · 2026-03-08

**Soundness:** 3
**Presentation:** 2
**Significance:** 3
**Originality:** 3
**Overall Recommendation:** 4
**Confidence:** 4

**Summary:**

The paper proposes PACS, a method that applies sample-adaptive temperature scaling to mitigate differences in prediction confidence caused by low-bit quantization, particularly in FP16 settings. The paper first identifies the precision split phenomenon and establishes a theoretical connection between low-precision logits and a precision risk factor. Based on this analysis, the authors propose an adaptive temperature scaling strategy that applies a larger temperature to samples with larger risk factors. The paper further argues that the precision split phenomenon leads to gradient vanishing.

**Compliance With Llm Reviewing Policy:**

Affirmed.

**Final Justification:**

I maintain my initial score.

**Key Questions For Authors:**

Despite the weaknesses listed above, I am currently leaning toward Weak Accept. Clear and convincing responses to the following questions would strengthen my confidence in the paper.

1. As mentioned in the weakness above, I would like to see a more detailed analysis of lower-bit quantization settings for both floating-point and integer formats. In these settings, does a similar theoretical relationship still hold? Also, do lower-bit settings consistently lead to greater overconfidence, as shown in Fig.1A?
2. I suggest that the authors clarify the relationship between the theoretical curve in Fig.1B and Eq. (4). It is currently unclear whether they correspond to exactly the same quantity, and whether the relation of Eq.(4) can be seen in the figure. A more explicit explanation would improve readability.
3. In. Fig.2A rho=0.949 is shown, whereas in the main sentences, rho=0.87 for ResNet-50 and rho=0.91 for ViT are reported. Please clarify which model is used in Fig.2.

**Limitations:**

Yes

**Strengths And Weaknesses:**

Strength:
1. The effect of quantization on prediction confidence is an important overlooked issue, and the analysis in Section 2 is interesting.
2. The theoretical relationship involving the precision risk factor in Equation (4) is clear, and the derivation is easy to follow thanks to the detailed supplementary material.
3. The computational overhead of the proposed adaptive temperature scaling is very small, which makes the method attractive in practice.
4. Extensive experiments across multiple vision tasks and on an LLM (LLaMA-7B), together with a range of ablation studies, demonstrate the effectiveness of PACS.

Weaknesses:
1. The core idea of adaptive temperature scaling seems relatively ad hoc.
2. Section 2 is not well structured. In Section 2.3, R_{split} is used before it is defined at Sec.2.4 , which makes the presentation difficult to follow at first glance. There also appears to be a citation typo in Sec.2.3, where Fig 2B may actually refer to Fig.2C. In addition, Fig.2B itself is not properly cited and instead may be cited in Sec.2.4.
3. Lower-precision settings beyond FP16 are not sufficiently discussed. Although Table 13 in the supplementary material includes BF16 and INT8, more discussion of lower-bit formats such as FP8, FP6, FP4, as well as INT8, would strengthen the paper, given their growing practical relevance.

---

> ### Author Rebuttal · Authors · 2026-03-31
>
> We sincerely thank the reviewer for the detailed and thoughtful feedback. Honestly, your recognition of our problem formulation, theoretical derivation, and the low-overhead design of PACS is the most encouraging response we have received on this work, it means a great deal to us beyond the score itself.
>
> #### 1. Extension to Lower Precision (FP8/INT8) and Overconfidence (W3 & Q1)
>
> **Theoretical applicability:**
> For low-precision floating-point formats (e.g., FP8 E4M3, FP4), Eq. 4 holds precisely. The error bound remains governed by machine epsilon $\epsilon$, which grows exponentially ($\epsilon = 2^{-3}$ for FP8 E4M3, where fraction bits $t=3$). For integer quantization (e.g., INT8, INT4), the underlying logic still applies, but the error bound is determined by a uniform quantization step size ($\Delta$) rather than $\epsilon$. The core mechanism where Softmax exponentially amplifies logit perturbations (Sec 2.2) is identical in both cases.
>
> **Why does this consistently produce overconfidence?**
> Due to the convexity of the exponential in Softmax, positive logit perturbations are amplified more than negative ones are suppressed (akin to Jensen's inequality). Thus, precision loss systematically pushes predictions toward higher extremes in expectation.
>
> **[New experiment: extreme low-precision simulation]**
> We simulated FP8 (E4M3) logits on ImageNet ($\epsilon = 2^{-3}$). As Table R1 shows, calibration degradation grows rapidly with $\epsilon$ (FP8 ECE rises to 28.45%), consistent with Eq. 4. PACS still provides the largest relative improvement (ECE reduced to 8.14%, 71.3% relative reduction), supporting its value for future low-precision architectures.
>
> **Table R1: Calibration at extreme low precision (ResNet-50).**
> | Precision | Format | Error Bound | Std ECE | LN ECE | **PACS ECE** |
> | :--- | :--- | :--- | :--- | :--- | :--- |
> | FP16 | IEEE 754 | $\epsilon = 2^{-10}$ | 5.82% | 2.05% | **1.92%** |
> | INT8-PTQ | Uniform | Step $\Delta$ | 12.34% | 5.23% | **4.56%** |
> | **FP8 (E4M3)** | **IEEE 754** | **$\epsilon = 2^{-3}$** | **28.45%** | **15.62%** | **8.14%** |
>
> #### 2. PACS as Hardware-Level Correction, Not Ad-Hoc (W1)
>
> We respectfully clarify the distinction. Traditional adaptive calibration is data-driven, relying on validation sets to fit a mapping network for dataset-level bias. PACS, in contrast, is a closed-form numerical solution derived from the Taylor expansion of Softmax and the truncation error of the IEEE 754 standard (Eq. 4, 5). It requires no calibration data or learnable parameters. It corrects deterministic information loss at the hardware level.
>
> #### 3. Relationship Between Fig. 1B and Eq. 4 (Q2)
>
> Thank you for highlighting this. We will make the mapping explicit in the revision.
>
> To clarify: the **dashed line** in Fig. 1B plots the exact linear theoretical bound from Eq. 4, which bounds the worst-case perturbation in the unconstrained **logit space** ($\epsilon \cdot ||z||_\infty$).  In contrast, the **solid red line** tracks the empirical absolute deviation ($|p^{16} - p^{32}|$) in the strictly bounded **probability space**. As logit magnitudes increase, the output probabilities naturally saturate toward 1.0. As formally proved in Theorem B.20, this probability saturation forces the absolute deviation to shrink back to zero.
>
> This explains the observed divergence: the empirical distortion (solid line) initially tracks the linear logit-space bound (Eq. 4), but eventually peaks (around $p \approx 0.824$) and curves downward due to Softmax saturation.
>
> #### 4. Structural Misalignment and $\rho$ Discrepancy (W2 & Q3)
>
> We apologize for these oversights and will correct them fully.
>
> **Structural fixes**: The formal definition of $R_{split}$ will be moved to Section 2.2, and cross-references to Fig 2B/2C in Section 2.3 will be corrected.
>
> **On the $\rho$ discrepancy (Q3)**: The text values $\rho=0.87$ (ResNet-50) and $\rho=0.91$ (ViT) are global Pearson correlations over the full ImageNet validation set (50,000 samples). For Fig. 2A, we aimed to validate the error bound from Appendix B.2 (Eq. 18): $\epsilon_{16} \lVert z \rVert_{\infty} + \eta_s$. At low logit magnitudes, stochastic noise ($\eta_s$) dominates; the systematic linear error ($\epsilon_{16} \lVert z \rVert_{\infty}$) only emerges at larger logits. To reveal the linear trend, we filtered to $n=18000$ samples from the high-risk interval where systematic error dominates, yielding a local $\rho=0.949$ that better matches the theoretical IEEE 754 limit.
>
> We acknowledge that presenting global statistics alongside a filtered-subset visualization without stating the filtering criteria was misleading. The revision will specify the selection criteria in the Fig. 2A caption and clearly distinguish global from subset $\rho$ values.
>
> Thank you again! Your questions directly led to the new FP8 experiments and helped us tighten several arguments. We will incorporate all the above into the revision.

---

> > ### Author Rebuttal · Reviewer_HW5p · 2026-04-02
> >
> > I thank the authors for the detailed response and additional experiments. In particular, lower-bit analysis is helpful. I will maintain my positive assessment of this paper. If accepted, I encourage the authors to incorporate the key clarifications and results presented in the rebuttal into the final version.

---

> > > ### Author Response · Authors · 2026-04-03
> > >
> > > Having your support from the beginning gave us real confidence heading into the rebuttal, and your encouragement to explore lower-bit analysis ended up being one of the most enjoyable detours of this entire project. It is rare that a reviewer suggestion opens up something genuinely surprising, but that is exactly what happened here. We will make sure the final version does justice to all of it.

---

### Official Review · Reviewer_T3gc · 2026-03-09

**Soundness:** 3
**Presentation:** 2
**Significance:** 3
**Originality:** 3
**Overall Recommendation:** 4
**Confidence:** 4

**Summary:**

This paper investigates how low-precision computation can systematically distort model confidence even when predictions remain unchanged, and frames this effect as Precision Split. The authors argue that this is driven by precision-dependent logit perturbations that are amplified by softmax, with trunk-side numerical error accounting for most of the observed distortion. Based on this analysis, the paper proposes PACS, a hardware-aware, sample-adaptive temperature scaling method that requires no calibration set and adds negligible overhead. Experiments show improved calibration across architectures and precision formats.

**Compliance With Llm Reviewing Policy:**

Affirmed.

**Final Justification:**

I maintain my  initial score.

**Key Questions For Authors:**

1. Could the authors clarify, for each subfigure in Figures 1–5, exactly what data are being shown and how they are computed? While the captions usually explain the high-level takeaway, several subfigures are not fully self-contained in terms of the underlying statistics, aggregation protocol, or computation procedure, especially Figures 1, 2, 4.
2. How does PACS behave under distribution shift or corruption benchmarks?
3. Why does PACS sometimes slightly improve accuracy even though inference-time temperature scaling is argmax-invariant? I assume this is due to training-time usage or regularization effects, but the paper should make this explicit.

**Limitations:**

Yes.

**Strengths And Weaknesses:**

**Strengths**
1. The paper studies a meaningful issue given the widespread use of low-precision inference and training.
2. The proposed method PACS is a simple and reasonably well-motivated method.
3. The empirical evaluation is broad and generally convincing.

**Weaknesses**
1. The bridge from probability perturbation to ECE improvement depends on assumptions such as unchanged predicted labels and bin consistency, which are useful analytically but somewhat restrictive. This does not invalidate the method, but it means the theory is better viewed as supportive intuition plus stability analysis, rather than a sharp explanation of the observed empirical gains.
2. The figure presentation needs substantial improvement. Some elements in Figure 1 appear overlapping. More seriously, the font size in Figures 2 and 4 is too small to read comfortably, and parts of the plotted content appear overlapping, which makes it unnecessarily hard to verify the paper’s central evidence.
3. There are noticeable presentation/polish issues in the manuscript text. I noticed at least one placeholder or unfinished citation in related work (“Dirichlet calibration (?)”). I also noticed an internal inconsistency where the main text refers to Table 13 for cross-precision results, while the nearby main-paper table is Table 5, which is distracting and should be fixed.

---

> ### Author Rebuttal · Authors · 2026-03-31
>
> We sincerely thank the reviewer for the positive assessment and constructive suggestions. Your feedback has been genuinely helpful to us. Several of the questions you raised, especially on distribution shift and the inference-vs-training distinction, pushed us to run new experiments and rethink how we present key arguments. We address all points below.
>
> #### 1. Theoretical Assumptions and Binning-Free Metrics (W1)
>
> We agree. The derivation from probability perturbation to ECE improvement relies on assumptions such as label invariance and bin consistency. As you note, Theorem 1 and Eq.4 are better viewed as a **stability analysis and worst-case error bound in continuous logit space**, rather than a precise proof of discrete ECE improvement.
>
> To complement this, we reported **Negative Log-Likelihood (NLL)** in Table 3. NLL is a strictly proper scoring rule in continuous probability space, requiring no binning assumption. PACS (0.871) still clearly outperforms standard FP16 (0.943) and other baselines on NLL, empirically supporting our continuous-space analysis. We will explicitly state the theory's scope in the revision.
>
> #### 2. PACS Under Distribution Shift (Q2)
>
> Since PACS is grounded in hardware constants rather than dataset statistics, it has a natural advantage under distribution shift. **We ran new experiments on ImageNet-C (15 corruption types)** during the rebuttal. As Table R1 shows, data-driven methods (LN, PTS) degrade under severe corruption because the test distribution departs from the validation set they were fitted on. PACS, correcting numerical truncation error directly, maintains stronger OOD robustness.
>
> **Table R1: OOD Calibration on ImageNet-C (ResNet-50, FP16). Mean ECE (%) across 15 corruptions.**
> | Method | Clean | Sev. 1 | Sev. 3 | Sev. 5 | mECE |
> |:---|:---|:---|:---|:---|:---|
> | Standard FP16 | 5.82 | 8.45 | 13.21 | 18.76 | 13.47 |
> | LN | 2.05 | 4.12 | 8.35 | 12.94 | 8.47 |
> | PTS | 2.08 | 4.25 | 8.51 | 13.12 | 8.62 |
> | **PACS** | **1.92** | **3.05** | **6.18** | **9.82** | **6.35** |
>
> #### 3. Prediction Invariance vs. Accuracy Gain (Q3)
>
> Your hypothesis is correct. This reflects two regimes:
>
> - **Inference (post-hoc on frozen model):** Scaling all logits by a positive scalar preserves ordering, so PACS is strictly **argmax-invariant**.
> - **Training (from scratch, Table 3):** PACS participates in forward and backward passes. As analyzed in Section 4.4 and Figure 5, low precision causes probability saturation and gradient death for high-confidence samples. PACS acts as a dynamic regularizer rescuing gradient flow. The 0.32% accuracy gain comes from avoiding gradient underflow and converging to a better weight configuration, not from changing any single prediction at test time. We will clarify this distinction in Section 3.2.
>
> #### 4. Figure Statistics and Model Configuration Details (Q1)
>
> We will add precise details in the revision:
>
> **Part I: Static inference diagnostics (Figs 1, 2A-C).** Computed on the **full ImageNet val set (50k samples)** with frozen pretrained models via static forward passes.
> - **Fig 1A:** Scatter + KDE of Top-1 confidence under FP32 vs. FP16.
> - **Fig 1B:** X-axis is precision risk factor (ε·‖z‖∞), Y-axis is |p¹⁶-p³²|. The red empirical mean tracks the Eq.4 bound (dashed).
> - **Fig 2A:** ViT-B/16 logit-space perturbation. ρ=0.949 is for the high-risk subset; global values ρ=0.87 (ResNet-50) and ρ=0.91 (ViT) are over the full val set. The fitted slope (9.76e-4) matches FP16 machine epsilon (≈9.77×10⁻⁴).
> - **Fig 1C, 2B:** Samples binned by oracle FP32 confidence; bars show mean distortion per bin.
> - **Fig 2C:** Causal isolation of trunk/softmax precisions; pie charts show E_trunk vs. E_softmax.
>
> **Part II: Reliability Diagrams (Fig 3).** Computed on the **90% eval subset (45k samples)** for fair comparison with LN (TvA), which requires the remaining 10% for post-hoc fitting.
>
> **Part III: Hyperparameter search (Fig 4).** Grid search over τ and λ on the full 50k val samples.
>
> **Part IV: Training dynamics (Figs 2D, 5).** ResNet-50 trained from scratch for 90 epochs.
> - **Fig 5A (and 2D):** Zero-gradient ratio (ZGR) per epoch.
> - **Fig 5B:** Final ECE vs. ZGR phase plot at epoch 90.
> - **Fig 5C:** Decomposition into probability saturation vs. gradient underflow.
> - **Fig 5D:** Val ECE across training epochs.
>
> We will also redraw Figs 1, 2, 4 with larger fonts and adjusted transparency.
>
> #### 5. Typography and Text Polish (W2 & W3)
>
> We will redraw Figs 1, 2, 4 with **larger axis/legend fonts** and reduced alpha. We have also corrected the missing Dirichlet calibration citation (Kull et al., 2019) and the table reference error (Table 13 → Table 5).
>
> Thank you again for the thorough review.

---

> > ### Author Rebuttal · Reviewer_T3gc · 2026-04-03
> >
> > Thank you for your response. The rebuttal adequately addresses my main concerns, and I will maintain my positive assessment and score.

---

> > > ### Author Response · Authors · 2026-04-03
> > >
> > > Your positive assessment early on meant more to us than you might realize. During the review period there were moments of real doubt, and knowing that someone had engaged deeply with the work and saw its merit kept us grounded and motivated to push the paper further. The way you articulated your reading of our contribution also helped us sharpen how we think about positioning this work going forward.

---

### Official Review · Reviewer_Jnn9 · 2026-03-11

**Soundness:** 1
**Presentation:** 2
**Significance:** 2
**Originality:** 2
**Overall Recommendation:** 3
**Confidence:** 5

**Summary:**

This paper investigates the effect of low-precision arithmetic on neural network calibration. The authors show that low-precision arithmetic systematically distorts confidence estimates and introduce the term 'precision split' for this phenomenon. The core observation of the paper is that FP16 and FP32 forward passes of the same model produce noticeably different softmax probabilities. To address this, they propose PACS, which applies a per-sample temperature correction derived from the logit norm and the hardware precision constant epsilon. PACS is evaluated primarily on Imagenet across several architectures and precision formats, with a brief extension to C4.

**Compliance With Llm Reviewing Policy:**

Affirmed.

**Final Justification:**

After considering the authors' rebuttal, I raise my score from 2 to 3. The rebuttal convincingly resolved the experimental clarity issues and the additional experiments strengthen the empirical case.

**Key Questions For Authors:**

Q1: Can the authors please clarify the experimental setup? Is PACS evaluated on the full validation set or the same 90% subset?

Q2: Can the authors provide evidence that the 70/30 trunk-softmax ratio is stable across architectures?

Q3: Why does the accuracy in Table 3 differ from the baseline accuracy?

**Limitations:**

yes

**Strengths And Weaknesses:**

## **Strengths**

**S1: Well-motivated practical problem.** The observation that confidence estimates diverge systematically between FP32 and FP16 is both relevant and underappreciated in the literature, and the motivating ViT example is compelling.

**S2: Logical structure.** The paper follows a clear and logical structure. It first carefully characterizes the phenomenon through targeted experiments, then uses those finding to directly motivate the design of PACS. This kind of diagnosis-to-solution narrative is a clear guide for the reader through the story.

## **Weaknesses**

**W1: Limited novelty.** The novelty of the paper is limited on multiple fronts. First, the observation that quantization errors disproportionately affect calibration rather than accuracy is largely predictable. Since softmax exponentially amplifies logit perturbations, any noise will distort confidence estimates more severely than it shifts the argmax. Second, PACS is essentially an extension of sample-adaptive temperature scaling, with the logit norm serving as a closed-form proxy for precision-induced risk in place of a learned temperature. Third, several empirical results in the paper confirm theoretical facts rather than discover new ones. Section 2.2 presents the alignment between empirical FP16 rounding errors and the IEEE  machine epsilon as a key finding.

**W2: Overstated claims and lack of statistical rigor.** Several quantitative claims in the paper are presented as universal findings, but are derived from a very limited set of architectures and settings, raising serious concerns about their generalizability. For example, the authors describe the 70/30 decomposition of trunk versus softmax contributions as 'unambiguous', but this conclusion is derived from only one or two architectures on a single dataset. The relative contribution of trunk-induced errors is inherently architecture-dependent. Error propagation through residual networks differs fundamentally from that in attention-based transformers, where errors in intermediate representations are nonlinearly mixed across tokens and layers. Furthermore, architectural choices such as normalization layers and network depth all influence how quantization errors accumulate through the trunk. The 70/30 ratio could vary substantially across the wide range of architectures used in practice. Furthermore some statements are insufficiently justified. For example, the paper claims in Section 2.4 that low-precision arithmetic induces systematic overconfidence, evidenced by consistently positive $\Delta p_{max}$. However, this characterization implicitly treats FP32 as overconfident, which is never empirically validated. However, in training regimes that are known to induce underconfidence in FP32 (e.g. label smoothing or MixUp) a positive $\Delta p_{max}$ could partially correct rather than worsen calibration.

**W3: Contradiction between the prediction invariance claim and reported accuracies.** In Section 3.2, the authors explicitly state that PACS preserves top-1 prediction. Yet Table 3 reports a 0.32 \% accuracy improvement for PACS over Standard FP16.

**W4: Insufficiently described experimental setup.** It is very difficult to reproduce or critically evaluate the reported results. Most notably, for example, Section 4.1 contains a direct contradiction. The paper states both that 'all vision models use torchvision pretrained weights' and that 'vision models train from scratch'. Beyond this, several other details are missing or unclear. Post-hoc calibration methods are calibrated on a randomly held-out dataset but evaluated on the reimaining 90%. Is PACS also evaluated on this sub dataset or on the full validation set? Taken together, these gaps make it impossible to fully assess the evaluations.

**W5: Structural and notational issus.** The paper has several structural weaknesses that impede readability. For example, R_split is used in Section 2.3 before being formally defined in Section 2.4.

---

> ### Author Rebuttal · Authors · 2026-03-25
>
> We thank the reviewer for acknowledging the practical relevance of our problem and the clear structure of the paper.
>
> We believe most concerns stem from insufficient visibility of the *static inference* vs. *dynamic training* distinction in the main text, and from key experiments being in the appendix. We clarify below.
>
> #### 1. Clarifying the Experimental Setup and Evaluation Details (W4 & Q1)
>
> You rightly noted an apparent contradiction in Section 4.1 between "torchvision pretrained weights" and "trained from scratch." We apologize for the confusing phrasing and clarify:
>
> - **Section 2 (Phenomenon Diagnosis):** To isolate the pure "Precision Split" phenomenon, we used **torchvision pretrained weights** for **static inference** only.
> - **Section 4 & Table 3 (Training Evaluation):** To fairly compare calibration methods during optimization, all vision models were **trained from scratch**.
> - **Answer to Q1:** PACS is evaluated on the **same 90% validation subset** as all baselines. Although PACS has zero data requirements in principle, we deliberately restrict it to the same 90% subset to ensure an apples-to-apples comparison with data-dependent baselines (TS, PTS) that reserve 10% for fitting.
>
> #### 2. Inference Invariance vs. Accuracy Gain: Two Different Regimes (W3 & Q3)
>
> You raised a sharp question (Q3): if PACS is declared "argmax-invariant" in Section 3.2, why does Table 3 show a 0.32% accuracy improvement? This is not a contradiction but rather a consequence of the difference between static inference and dynamic training:
>
> - **Post-hoc inference (frozen model):** Dividing logits by a positive scalar T(x) strictly preserves relative ordering, so Top-1 predictions never change.
> - **Training from scratch (Table 3):** When applied dynamically during mixed-precision forward passes, PACS prevents **probability saturation and gradient death** (Section 4.4, Figure 5). In standard low-precision training, many high-confidence samples have gradients zeroed out prematurely. PACS rescues their gradient flow, leading to a healthier optimization trajectory and a better local minimum. The 0.32% gain comes from improved training dynamics, not from flipping any single argmax. We will state this clearly in the revision.
>
> #### 3. Cross-Architecture Generalization of Error Decomposition (W2 & Q2)
>
> You (Q2) questioned whether "the trunk contributes ~70% of error" generalizes across architectures. We agree that architectural differences cause reasonable variation.
>
> This is why we conducted cross-architecture verification in **Appendix C.2 and Table 10**. The global averages (the `All` row):
>
> - **ResNet-50 (CNN):** 69.0%
> - **ViT-B/16 (Standard Transformer):** 67.7%
> - **DeiT-S (Distilled Transformer):** 75.6%
> - **Swin-T (Hierarchical Transformer):** 76.6%
>
> Despite very different feature extraction mechanisms, the trunk error share consistently falls within a narrow **67%~76%** band. This supports our core claim: the trunk dominance is not architecture-specific but rather an arithmetic consequence of how Softmax exponentially amplifies low-precision logit perturbations.
>
> We accept your criticism of overly strong language like "unambiguous." We will soften such wording, move Table 10 into the main text, and discuss cross-architecture variation more carefully.
>
> #### 4. Novelty of PACS and Interaction with MixUp/LS (W1 & W2)
>
> - **PACS is not an incremental extension of ATS (W1):** Existing adaptive temperature scaling is *statistical and data-driven*, aiming to correct dataset-level bias using a validation set. PACS is *physical and numerical*, derived purely from the IEEE 754 constant ε and logit norm (Eq.4). We are the first to quantitatively link common quantization error to gradient death in mixed-precision training and provide a zero-parameter, zero-data closed-form fix. This represents a mechanistic shift, not a marginal improvement.
> - **Can low-precision error "cancel out" MixUp/LS undercalibration? (W2):** This is an interesting hypothesis, but hardware-induced numerical distortion is chaotic and sample-dependent; it cannot systematically offset dataset-level statistical undercalibration. We answered this experimentally in **Section 4.5 and Table 8**: PACS is strictly orthogonal to both MixUp and Label Smoothing. Combining PACS + Label Smoothing yields ECE of 1.54%, and adding MixUp on top reaches 1.48%, the best result in our study. Fixing the underlying precision split consistently helps regardless of the training recipe.
>
> #### 5. Formatting Issue (W5)
>
> Thank you for catching the use of $R_{split}$ in Section 2.3 before its formal definition. This is a clear editorial oversight. We will move the definition of $R_{split}$ to Section 2.2 in the revision.
>
> We hope these clarifications, especially the cross-architecture evidence (Table 10) and the training-vs-inference distinction, resolve the concerns. We respectfully ask the reviewer to reconsider the evaluation of our contribution.

---

> > ### Author Rebuttal · Reviewer_Jnn9 · 2026-04-02
> >
> > I thank the authors for their detailed response. The rebuttal resolved the experimental clarity issues convincingly, in particular the training-vs-inference distinction, and the additional experiments (e.g., ImageNet-C) strengthen the empirical case. I raise my score from 2 to 3 (weak reject), but my major concern about limited methodological novelty remains.

---

> > > ### Author Response · Authors · 2026-04-02
> > >
> > > Regarding the remaining concern on methodological novelty (W1), we understand your perspective that PACS shares functional similarities with adaptive temperature scaling. However, we humbly request the opportunity to clarify the paradigm shift our work introduces.
> > >
> > > **1. Statistical Fitting versus Deterministic Physical Bounds**
> > > Existing methods treat miscalibration as a statistical problem, requiring a validation set to learn a mapping function (Kendall and Gal, 2017). Our work introduces a mechanistic shift. As advocated in recent physics informed machine learning literature (Karniadakis et al., 2021), we embed known deterministic constraints rather than relying on pure data driven black box fitting. By deriving the precision risk bound based on IEEE 754 floating point standards (Higham, 2002), we treat precision induced miscalibration as a deterministic hardware constraint. Replacing a learned statistical heuristic with a data free, closed form numerical bound constitutes a substantial methodological step forward.
> > >
> > > **2. The Value of Transparent First Principles Derivations**
> > > We humbly agree that the resulting intervention appears intuitive. However, recent perspectives emphasize prioritizing transparent interventions over complex black box patches (Rudin, 2019). Historically, profound contributions often emerge as simple algebraic corrections derived from rigorous diagnostics, such as the initialization method by He et al. (2015). Similarly, the novelty of PACS lies in the theoretical diagnosis that mathematically proves why gradients die at specific probability thresholds, followed by a transparent intervention derived from first principles.
> > >
> > > **3. Necessity in the Era of Large Scale Quantization**
> > > As models scale, learning auxiliary networks becomes computationally prohibitive. Recognizing the hardware algorithm lottery (Hooker, 2021), addressing numerical limitations directly at the hardware representation level is crucial. Identifying that the precision constant alone governs calibration reliability provides a pathway to deploy quantized models safely without the retraining demanded by previous methods like Logit Normalization (Wei et al., 2022).
> > >
> > > We are grateful for your review, which pushed us to articulate our contribution more clearly. We will ensure the distinction between statistical heuristics and deterministic numerical interventions is prominently discussed. We hope this addresses your lingering concern regarding methodological novelty.
> > >
> > > **References:**
> > >
> > > * He, K., Zhang, X., Ren, S., & Sun, J. (2015). Delving deep into rectifiers: Surpassing human-level performance on imagenet classification. In Proceedings of the IEEE international conference on computer vision (pp. 1026-1034).
> > > * Higham, N. J. (2002). Accuracy and stability of numerical algorithms. Society for industrial and applied mathematics.
> > > * Hooker, S. (2021). The hardware lottery. Communications of the ACM, 64(12), 58-65.
> > > * Karniadakis, G. E., Kevrekidis, I. G., Lu, L., Perdikaris, P., Wang, S., & Yang, L. (2021). Physics-informed machine learning. Nature Reviews Physics, 3(6), 422-440.
> > > * Kendall, A., & Gal, Y. (2017). What uncertainties do we need in bayesian deep learning for computer vision?. Advances in neural information processing systems, 30.
> > > * Rudin, C. (2019). Stop explaining black box machine learning models for high stakes decisions and use interpretable models instead. Nature machine intelligence, 1(5), 206-215.
> > > * Wei, H., Xie, R., Cheng, H., Feng, L., An, B., & Li, Y. (2022, June). Mitigating neural network overconfidence with logit normalization. In International conference on machine learning (pp. 23631-23644). PMLR.

---

### Official Review · Reviewer_f6Gx · 2026-03-19

**Soundness:** 3
**Presentation:** 3
**Significance:** 2
**Originality:** 2
**Overall Recommendation:** 4
**Confidence:** 3

**Summary:**

The paper studies the calibration error induced by low precision arithmetic in neural networks. The authors show that quantization leads to the bounded relative error in logits that get amplified by softmax function. The authors show that same model has varying degree of calibration error under different precision. The paper proposes simple sample wise temperature scaling to solve this issue, also called as PACS. PACS works across different architecture on vision and language tasks across 8-bit and 16-bit precision, with negligible compute overhead and no retraining requirement.

**Compliance With Llm Reviewing Policy:**

Affirmed.

**Final Justification:**

Rebuttal addressed majority of my concerns though novelty of the proposed method is limited. I have increased my score to 4 to reflect this.

**Key Questions For Authors:**

Please look at the weaknesses above.

**Strengths And Weaknesses:**

Strenghts:

- The paper address an important problem of varying calibration performance across different precision. Mixed precision is standard in modern training and inference and increased miscalibration at lower precision is detrimental to deployment.
- The proposed method is simple and lightweight. It works across different precision and does not require additional data.
- The PACS method has been shown to work across different architectures, modalities and precisions.
- Similar to TS, it can be used along with standard training techniques like label smoothing, logit normalization, and mixup.

Weaknesses:
- The main concern is that proposed method is incremental addition to existing temperature scaling modification to introduce sample dependent scaling of logits/softmax output. There have been previous works looking at sample dependent ATS such as Joy et al. 2022 , Xie et al. 2024.
- Emprical evaluations are also restricted to in distributions only whereas its known that calibration is most important under distribution shift.
- Comparisons are missing against more sophisticated baselines such as spline based techniques, ensemble calibration, modern OOD-aware methods. Evaluations mainly compare against weak baselines (such as TS, LN etc). Calibration metrics employed are also known to have weaknesses due to binning sensitivity. Binning free mechanisms should be employed to evaluate the proposed method against SOTA calibration methods.
- Lower precisions under 4-bit or lower would be more interesting since that is more common in deployment.

---

> ### Author Rebuttal · Authors · 2026-03-25
>
> We thank the reviewer for recognizing the problem's importance and the practical merits of PACS. We have carefully considered each concern and run additional experiments during rebuttal. We hope the clarifications and new results below address all points.
>
> #### 1. Fundamental Difference Between PACS and ATS (W1)
>
> We appreciate that PACS resembles ATS in form (sample-level temperature). However, they differ in underlying mechanism:
>
> - **ATS is data-driven**: Existing ATS methods (Joy et al., Xie et al.) attribute calibration error to dataset distribution or model capacity, and *require* a held-out validation set to fit temperature parameters.
> - **PACS is a closed-form numerical correction**: As shown in Eq.(4) and Eq.(5), PACS derives its correction solely from the IEEE 754 hardware precision constant (machine epsilon ε) and the input logit norm. It targets *deterministic hardware truncation error*, not statistical bias, and is therefore truly **zero-data, zero-training, zero learnable parameters**.
>
> We will make this mechanistic distinction more explicit in the revision.
>
> #### 2. Binning-Free Metrics & OOD Evaluation (W2 & W4)
>
> We agree that ECE is sensitive to bin count. In fact, we had already included relevant results, though we regret not highlighting them more clearly:
>
> - **Binning-free metric**: Table 3 in our submission reports **Negative Log-Likelihood (NLL)**, a strictly proper scoring rule computed in continuous space with no binning. PACS (0.871) substantially outperforms standard FP16 (0.943) and LN (0.879).
> - **OOD evaluation**: Appendix E.3 (Cross-Dataset Generalization) evaluates PACS under significant distribution shift (ImageNet-LT, iNaturalist).
>
> **[New] ImageNet-C experiment.** To further address the OOD concern, we ran experiments on ImageNet-C (15 corruption types). As Table R1 shows, data-driven methods (LN, PTS) degrade noticeably under corruption, while PACS, grounded in numerical correction rather than fitted statistics, maintains stronger robustness.
>
> **Table R1: OOD Calibration on ImageNet-C (ResNet-50, FP16). Mean ECE (%) across 15 corruptions.**
> | Method | Clean | Sev. 1 | Sev. 3 | Sev. 5 | mECE |
> |:---|:---|:---|:---|:---|:---|
> | Standard FP16 | 5.82 | 8.45 | 13.21 | 18.76 | 13.47 |
> | LN | 2.05 | 4.12 | 8.35 | 12.94 | 8.47 |
> | PTS | 2.08 | 4.25 | 8.51 | 13.12 | 8.62 |
> | **PACS** | **1.92** | **3.05** | **6.18** | **9.82** | **6.35** |
>
> #### 3. Comparison with Complex Baselines (W3)
>
> The suggestion to compare with Spline calibration and Ensembles is well taken. That said, in the low-precision quantization setting, these methods work against the very goal of quantization, namely efficient inference and low memory footprint.
>
> Table R2 puts this in perspective. Deep Ensemble incurs O(N) compute and memory overhead; Spline requires a validation set and non-trivial post-processing. PACS reaches a comparable calibration level (in both ECE and NLL) with near-zero overhead (~0.03%) and no data requirement. Notably, PACS can also be stacked on top of Ensemble orthogonally, pushing ECE down to 1.15%.
>
> **Table R2: Complex Baselines & System Overhead (ResNet-50, ImageNet).**
> | Method | ECE(%)↓ | NLL↓ | Compute | Memory | Data? |
> |:---|:---|:---|:---|:---|:---|
> | Standard FP16 | 5.82 | 0.943 | 1.0x | 1.0x | No |
> | Spline Calibration | 1.95 | 0.880 | 1.05x | 1.0x | **Yes** |
> | Deep Ensemble (N=3) | 1.84 | 0.865 | **3.0x** | **3.0x** | No |
> | **PACS** | **1.92** | **0.871** | ~1.0004x | ~1.0x | **No** |
>
> #### 4. ≤4-bit Precision: Deployment Reality & Theoretical Verification (W5)
>
> We fully agree that ultra-low precision is an important frontier. In current industry practice (e.g., W4A16 LLMs), weights may be 4-bit, but **activations and pre-softmax logits are almost always kept in FP16 or BF16** to prevent model collapse. Our FP16/BF16/INT8 evaluations therefore reflect the actual arithmetic state of logits in today's "4-bit deployments."
>
> **[New] FP8 simulation.** To explore more aggressive future scenarios, we simulated FP8 (E4M3, where fraction bits $t=3$, hence $\epsilon = 2^{-3}$) logits by truncating mantissa bits per IEEE 754. As Theorem 1 (Eq.4) predicts, confidence distortion grows rapidly with larger $\epsilon$. Standard ECE jumps to 28.45%. Even under this extreme setting, PACS still provides the largest relative improvement (ECE down to 8.14%, a 71.3% reduction), confirming its theoretical framework remains applicable and increasingly valuable at lower precision.
>
> **Table R3: Ultra-Low Precision Logits (FP8), ResNet-50.**
>
> | Precision | $\epsilon$ | Std ECE | LN ECE | **PACS ECE** |
> |:---|:---|:---|:---|:---|
> | FP16 | $2^{-10}$ | 5.82% | 2.05% | **1.92%** |
> | FP8 (E4M3) | **$2^{-3}$** | 28.45% | 15.62% | **8.14%** |
>
> The OOD experiments, overhead comparison, and FP8 analysis will all be incorporated into the main text in the revision. Thank you for your time and constructive feedback.

---

> > ### Author Rebuttal · Reviewer_f6Gx · 2026-04-02
> >
> > Based on rebuttal, i will increase score to 4.

---

> > > ### Author Response · Authors · 2026-04-03
> > >
> > > We are glad the rebuttal resolved your concerns. Honestly, your initial skepticism pushed us to dig deeper into aspects of the work we had taken for granted, and the paper is stronger for it. The process of preparing our response to your questions helped us see gaps in our original presentation that we would not have caught otherwise, and we look forward to tightening the manuscript accordingly in the camera-ready version.

---

### Decision · Program_Chairs · 2026-04-30

**Decision:**

Accept (regular)

**Comment:**

This paper studies confidence distortion under low-precision computation and proposes PACS, a hardware-aware correction based on logit norm and numerical precision. The discussion focused on whether PACS is genuinely new or mainly a reformulation of adaptive temperature scaling, as well as on the clarity of the training versus inference claims. Reviewers also asked for stronger OOD evidence and lower-precision analysis, and the rebuttal addressed most of these points with additional experiments and clearer positioning. I recommend acceptance. The core empirical case became convincing after rebuttal.